# Evaluating In-Sample Softmax in Offline Reinforcement Learning: An Analysis Across Diverse Environments

## Abstract

In this work, we considered the problem of learning action-values and corresponding policies from a fixed batch of data. The algorithms designed for this setting need to account for the fact that the action-coverage of the data distribution may be incomplete, that is certain state-action transitions are not present in the dataset. The core issue faced by Offline RL methods is insufficient action-coverage which leads to overestimation or divergence in learning during the bootstrapping update. We critically examine the In-Sample Softmax (INAC) (Xiao et al., 2023) algorithm for Offline Reinforcement Learning (RL), addressing the challenge of learning effective policies from pre-collected data without further environmental interaction using an in-sample softmax. Through extensive analysis and comparison with other in-sample algorithms like In-sample Actor-Critic (IAC) (Zhang et al., 2023) and Batch-Constrained Q-learning (BCQ) (Fujimoto et al., 2019), we investigate INAC's efficacy across various environments, including tabular, continuous, and discrete domains, as well as imbalanced datasets. We find that the INAC, when benchmarked against state-of-the-art offline RL algorithms, demonstrates robustness to variations in data distribution and performs comparably, if not superiorly, in all scenarios. We do a comprehensive evaluation of the capabilities and the limitations of the In-Sample Softmax method within the broader context of offline reinforcement learning.

## 1 Introduction

Offline reinforcement learning (offline RL) considers the problem of learning effective policies entirely from previously collected data. This is very appealing in a range of real-world domains, from robotics to logistics and operations research real-world exploration with untrained policies is costly or dangerous, but prior data is available. This data could have been gathered under a near-optimal behaviour policy, from a mediocre policy, or a mixture of different policies (perhaps produced by several expert agents). One of the core challenges of offline RL is development of algorithms which perform well over various compositions of data distributions i.e they are able to learn the best actions, even if the number of such transitions in the dataset are minimal. The other more costly alternative to make offline RL perform well is to control the data collection to ensure that good trajectories are collected. Most methods for learning action values in Offline Reinforcement Learning (Offline RL) either use actor-critic algorithms, where the action values are updated using temporal-difference (TD) learning updates to evaluate the actor, or Q-learning updates, which bootstrap off of a maximal action in the next state. Regardless in both approaches, insufficient action coverage can lead to poor performance when combined with bootstrapping.

The action-value updates based on TD involve bootstrapping off an estimate of values in the next state. This bootstrapping is problematic if the value is overestimated, which is likely to occur when actions are never sampled in a state (Fujimoto et al., 2018b; Kumar et al., 2019; Fujimoto et al., 2019). This overestimated action will be selected when using a maximum over actions, pushing up the value of the current state and action. Such updates can lead to poor policies and instability. Nevertheless, this imposes a trade-off between how much the policy improves and how vulnerable it is to misestimation due to distributional shifts. To solve the problems that arise due to the extrapolation error, state of the art algorithms such as Implicit Q-Learning (IQL) (Kostrikov et al. (2021)), In-sample Actor-Critic (INAC) (Xiao et al. (2023)) and In-sample

Actor-Critic (IAC) (Zhang et al. (2023)) use in-sample learning, where they attempt to learn an optimal policy, without ever querying the values of unseen actions.

In this work, we

1. Successfully reproduce the claims made for INAC in Xiao et al. (2023) and explored the extension of its adaptability to sub-optimal imbalanced datasets

2. Compare it with adaptations of IQL and CQL that use Return Weighting or Advantage Weighting sampling strategies (Hong et al., 2023; Peng et al., 2019)

3. Compare it to Batch Conservative Q-Learning (BCQ). This allows us to quantify the impact of using a soft-max over a maximum (as in INAC) versus using argmax over actions (as in BCQ)

4. Demonstrate that adding Behavioral Cloning regularization into the Actor loss function induces stability during training while keeping the mean reward the same

In summary, this reproduction contributes empirical results on the ability of In-Sample softmax to generalise to varied compositions of data distributions and provides results of rigorous experimentation to suggest INAC as a go-to algorithm for offline RL settings, especially when the quality of the dataset is unknown. This study replicated the experiments of the original authors of INAC (Xiao et al., 2023) whilst conducting further experimentation on imbalanced datasets (Hong et al., 2023) and including Behaviour Cloning Regularization (Fujimoto & Gu, 2021).

## 2 Background

### 2.1 Markov Decision Process

Let us define a finite Markov Decision Process (MDP) $M = (\mathcal{S}, \mathcal{A}, P, r, \gamma)$ (Puterman, 1994), where $\mathcal{S}$ is a finite state space, $\mathcal{A}$ is a finite action space, $\gamma \in [0, 1)$ is the discount factor, $r : \mathcal{S} \times \mathcal{A} \to \mathbb{R}$ and $P : \mathcal{S} \times \mathcal{A} \to \Delta(\mathcal{S})$ are the reward and transition functions. A *value function* is defined as the expected cumulative future rewards that an agent can obtain from a given state obtained by following a policy $\pi : \mathcal{S} \to \Delta(\mathcal{A})$, $v^\pi(s) = \mathbb{E}^\pi[\sum_{t=0}^\infty \gamma^t r(s_t, a_t)|s_0 = s]$ where $\mathbb{E}^\pi$ denotes the expectation under the distribution induced by the policy $\pi$ acting on the environment. The corresponding *action-value* or *Q-* function is $q^\pi(s, a) = r(s, a) + \gamma \mathbb{E}'_s \sim P(\cdot|s, a)[v^\pi(s')]$. An optimal policy $\pi^*$ exists that maximizes the values for all states $s \in S$. We use $v^*$ and $q^*$ to denote the optimal value functions. The optimal value satisfies the *Bellman optimality equation,*

$$v^*(s) = \max_a \left\{ r(s, a) + \gamma \mathbb{E}_{s'}[v^*(s')] \right\}, \quad q^*(s, a) = r(s, a) + \gamma \mathbb{E}_{s' \sim P(\cdot|s, a)}[\max_{a'} q^*(s', a')] \tag{1}$$

In InAC, we consider the *entropy-regularized MDP* setting—also called the maximum entropy setting—where an entropy term is added to the reward to regularize the policy towards being stochastic (Snoswell et al., 2020). The maximum-entropy value function is defined as

$$\widetilde{v^\pi}(s) = v^\pi(s) + \tau \mathbb{H}(s, \pi),$$

$$\text{where } \mathbb{H}(s, \pi) = \mathbb{E}_\pi \left[ \sum_{t=0}^\infty -\gamma^t \log \pi(a|s)|s_0 = s \right]$$

for temperature $\tau$. $\mathbb{H}$ is called the *discounted entropy regularization*. The corresponding maximum-entropy Q-function is $\widetilde{q^\pi}(s, a) = r(s, a) + \gamma \mathbb{E}_{s' \sim P(s, a)}[\widetilde{v}^\pi(s')]$. As $\tau \to 0$, we recover the original value function definitions.

## 2.2 Offline Reinforcement learning

Using offline reinforcement learning, we deal with the problem of learning an optimal decision making policy from a previously collected offline dataset $\mathcal{D} = \{s_i, a_i, r_i, s'_i\}_{i=0}^{n-1}$. The data is assumed to be generated by executing a *behavior policy* $\pi_{\mathcal{D}}$. In offline RL, only samples in this $\mathcal{D}$ can be used for learning, without further interaction with the environment. Offline RL algorithms do not have access to the full coverage of the action distribution, which leads to over-estimation of the value function. The policy learnt is prone to distribution shifts and yields poor results during test time. In order to overcome this issue, one approach that is implemented frequently is to constrain the learned policy to be similar to $\pi_{\mathcal{D}}$, like by adding a KL-divergence regularization term:

$$\max_{\pi} \mathbb{E}_{s \sim \rho} \left[ \sum_a \pi(a|s) q(s, a) - \tau D_{\mathrm{KL}}(\pi(.|s) || \pi_D(.|s)) \right] \tag{2}$$

for some $\tau > 0$. The optimal policy for this objective must be on the support of $\pi_{\mathcal{D}}$: the KL constraint makes sure $\pi(\text{a|s}) = 0$ as long as $\pi_{\mathcal{D}}(\text{a|s}) = 0$. This constraint can result in poor $\pi$' when $\pi_{\mathcal{D}}$ is sub-optimal, as confirmed in previous studies (Kostrikov et al., 2021).

The strategy that InAC serves to emulate involves considering an in-sample policy optimization,

$$\max_{\pi \leq \pi_{\mathcal{D}}} \pi(a|s) q(s, a), \pi \leq \pi_{\mathcal{D}} \forall a \in A \tag{3}$$

where $\pi \leq \pi_{\mathcal{D}}$ indicates the support of $\pi$ is a subset of $\pi_{\mathcal{D}}$. This approach more directly avoids selecting out-of-distribution actions. The idea is to estimate $\pi_\omega \approx \pi_{\mathcal{D}}$ and directly constrain the support by sampling candidate actions from $\pi_\omega$, just as it is proposed for Batch-Constrained Q-learning (Fujimoto et al., 2018a).

As far as out-of-sample actions are concerned, the in-sample policy optimization more explicitly constraints the support of policy to be a subset of the behavior policy $\pi_D$. This provides an advantage over the KL-divergence regularization, since, this merely makes the policy similar to the behavior policy without much constraint, that is, only using a regularization term in the objective.

## 2.3 In-Sample Softmax (INAC)

The in-sample AC algorithm (Xiao et al., 2023) learns an actor $\pi_\psi$ with parameters $\psi$, action-values $q_\theta$ with parameters $\theta$ and a value function $v_\phi$ with parameters $\phi$. Additionally, INAC learns $\pi_\omega \approx \pi_D$. INAC initially extracts a greedy policy $\pi_\omega \approx \pi_D$ using a simple maximum likelihood loss on the dataset: $L_{\mathrm{behavior}}(\omega) = -\mathbb{E}_{(s,a) \sim D}[\log \pi_\omega(a|s)]$. It will only be used to adjust the greedy policy and will only be queried on actions in the dataset. INAC alternates between estimating $q_\theta$ and $v_\phi$ for the current policy and improving the policy by minimizing a KL-divergence to the soft greedy policy. It updates its policy to attempt to match the in-sample soft greedy policy. We highlight this equation to signify the target policy we approach and its relation to the implicit policy $\pi_D$ and its approximation $\pi_\omega$. The derivation from hard-max to soft-max terms and the in-sample soft greedy policy is provided in (Xiao et al., 2023).

$$\hat{\pi}_{\pi_{\mathcal{D}}, q_\theta}(a \mid s) = \pi_{\mathcal{D}}(a \mid s) \exp \left( \frac{q_\theta(s, a) - Z(s)}{\tau} - \log \pi_\omega(a \mid s) \right) \tag{4}$$

where $Z(s) = \tau \log \int_a \pi_{\mathcal{D}}(a \mid s) \exp \left( \frac{q_\theta(s,a)}{\tau'} - \log \pi_\omega(a \mid s) \right) da$ is the normalizer. The final loss function for the actor $\pi_\psi$ is

$$\mathcal{L}_{\mathrm{actor}}(\psi) = -\mathbb{E}_{s, a \sim \mathcal{D}} \left[ \exp \left( \frac{q_\theta(s, a) - v_\phi(s)}{\tau} - \log \pi_\omega(a \mid s) \right) \log \pi_\psi(a \mid s) \right] \tag{5}$$

For the value function, we use standard value function updates for the entropy-regularized setting. The objectives are

$$\mathcal{L}_{\mathrm{baseline}}(\phi) = \mathbb{E}_{s \sim \mathcal{D}, a \sim \pi_\psi(s)} \left[ \frac{1}{2} \left( v_\phi(s) - (q_\theta(s, a) - \tau \log \pi_\psi(a \mid s)) \right)^2 \right] \tag{6}$$

$$\mathcal{L}_{\text{critic}}(\theta) = \mathbb{E}_{s,a,r,s'\sim\mathcal{D}}\left[\frac{1}{2}\left(r + \gamma v_\phi\left(s'\right) - q_\theta(s,a)\right)^2\right] \tag{7}$$

The action-values use the estimate of $v_\phi$ in the next state. Therefore, they do not use out-of-distribution actions. The update to the value function, $v_\phi$, uses only actions sampled from $\pi_\psi$, which is being optimized to stay in-sample. The actor update progressively reduces the probability of these out-of-distribution actions, even if temporarily the action-values overestimate their value because the actor update pushes $\pi_\psi$ towards the in-sample greedy policy.

## 2.4 Preliminaries

### 2.4.1 Batch-Constrained deep Q-learning (BCQ)

The algorithm Batch-Constrained deep Q-learning (BCQ) attempts to perform in-sample learning by employing a state generative model that produces only previously seen actions. It combines this with a deep Q-network to select the value with the highest action, similar to the batch data. Batch-constrained reinforcement Learning works by defining trust regions (Schulman et al., 2017a) in the batch and determining the Q-value only in these regions.

The state-conditioned generative model is used to generate actions likely under the given batch, which is then combined with a network to optimally perturb actions in a small range. The perturbation model increases the diversity of seen actions and prevents excessive prohibitive sampling from the generative model. A Q-network is used to obtain the highest valued action, and a duo of Q-networks are trained such that the value update penalizes unfamiliar states and rewards familiar ones. This value estimate bias is achieved through a modification to Clipped Double Q-learning (Fujimoto et al., 2018b). BCQ works by taking an in-sample max.

### 2.4.2 In-sample Actor Critic (IAC)

In-sample Actor Critic (IAC) (Zhang et al., 2023) performs iterative policy iteration and simultaneously follows in-sample learning to eliminate extrapolation error. It uses sampling-importance resampling to reduce variance and executes in-sample policy evaluation, which formulates the gradient as it is sampled from the trained policy. Sampling is done in accordance with the weights of importance resampling. For policy improvement, advantage-weighted regression (Peng et al., 2019) has been used to control the deviation from the behavior policy.

IAC executes unbiased policy evaluation and has a smaller variance than importance sampling. It leverages the benefits of both multi-step dynamic programming and in-sample learning, which only relies on the target Q-values of the actions in the dataset. Moreover, it is unbiased and has a smaller variance than importance sampling. In contrast to prior approaches, IAC dynamically adjusts the distribution of the dataset to align with the learned policy throughout the learning process.

### 2.4.3 Implicit Q-Learning

Implicit Q-Learning (IQL) (Kostrikov et al., 2021) is an offline RL method that aims to avoid querying out-of-sample actions in the TD loss. It approximates the policy improvement step implicitly by treating the state value function as a random variable. The algorithm never needs to evaluate unseen actions and the actions determine the randomness of the random variables. By taking a state conditional upper expectile of this random variable, the algorithm estimates the best actions in that state. It considers a fitted Q evaluation with a SARSA-like objective, which aims to learn the value of the dataset policy $\pi_B$.

### 2.4.4 TD3+BC

Fujimoto & Gu (2021) proposed that by adding a behaviour cloning term to the policy update of the TD3 algorithm (Fujimoto et al., 2018b), and then normalizing the data, one can match the performance of state of the art offline RL algorithms. They named this approach as TD3+BC. Adding a behaviour regularization term pushes policy towards favouring actions contained in the dataset.

# 3 Investigation and Results

We begin the investigation by reproducing the results shown in the paper (Xiao et al., 2023). The experiments performed and reported in the paper belong to the following broad categories:

1. Tabular Domain

2. Continuous Control MuJoCo environments

3. Discrete Control MuJoCo environments

and their details will be noted further in this section. We further introduce experiments on imbalanced datasets (Hong et al., 2023) generated and derived from D4RL (Fu et al., 2021) datasets and benchmark the performance of INAC on some state-of-the-art in-sample offline reinforcement learning algorithms and on standard state-of-the-art offline reinforcement learning algorithms.

## 3.1 Reproducibility

### 3.1.1 Tabular Domain

We first perform a sanity check on the tabular domain, a four-rooms custom grid-world environment. This would mean that INAC algorithm converges to the same policy as value iteration performed on the same environment. Since value iteration converges to an optimal policy, we use this as an Oracle prediction (i.e. a skyline). We denote this as Oracle-Max. In these experiments, we follow the paper's implementation of the custom 13 x 13 Four Rooms environment, where the agent starts from the bottom-left and needs to navigate through the four rooms to reach the goal in the up-right corner in as few steps as possible. There are four actions: A = {up, down, right, left}. The reward is zero on each time step until the agent reaches the goal-state where it receives +1. Episodes are terminated after 100 steps, and $\gamma$ is 0.9. We use three different behavior policies to collect three datasets from this environment called Expert, Random, and Missing-Action. The expert dataset contains data from the optimal policy generated by value iteration. In random dataset, the behavior policy takes each action with equal probability. The mixed dataset contains 10 percent of the data from the expert dataset and 90 percent of the data from the random dataset, which mirrors data generated by a sub-optimal policy. For the Missing-Action dataset, we remove all transitions taking down actions in the entire room from the Mixed dataset. In the Expert, Mixed and Random datasets we produce results that align with those given in the paper. In the Missing-Action dataset, where the entirety of the downward actions were removed, the algorithm did not reach the optimal value but nevertheless performed relatively well to give a normalized score of about 0.4 on a scale between 0 and 1. The results are displayed in Figure 1.

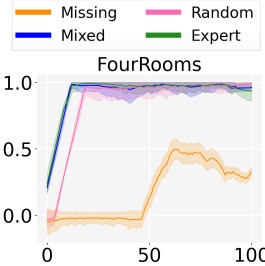

Figure 1: Average Returns vs Iterations (x$10^2$). Performance of INAC over four different datasets in tabular domain graphed as a sanity check. The algorithm converges to 1 (highest return) for all datasets except "missing-action".

### 3.1.2 Continuous environments

In continuous control tasks, we use the datasets provided by D4RL. We train the agent for 1 million iterations. We run Walker2D, Ant, Hopper, HalfCheetah on four datasets, Medium, Medium Expert, Expert, Medium

Replay using the corresponding optimal hyperparameters. The reproduced results on most of the continuous control environments are either better than or comparable to the results given in the paper for the optimal hyperparameters. We run a grid search on the learning rate and the Lagrangian factor $\tau$ for each environment. We verify that the prescribed hyperparameters are in fact, the optimal hyperparameters in the sweep. The hyperparameter sweep results are shown in the Appendix, Figure 5. The average returns, both normalized and unnormalized are displayed in Table 7 and Table 6. Note that the column named as INAC (Ours) refers to the INAC runs that we conducted, whereas the column named INAC refers to the runs that Xiao et al. (2023) conducted, which they averaged over 5 seeds. Similarly, the IAC values reported are the same as the ones reported by Zhang et al. (2023).

Table 1: The final performance of each algorithm in continuous action space environments for expert, medium expert, medium replay and medium datasets. Scores are normalized and averaged over 6 random seeds.

| Environment | Dataset | INAC (Ours) | INAC | IQL | IAC |
|---|---|---|---|---|---|
| **Ant** | Expert | **126.7** | 128.4 | 118.9 | NA |
| | Medium expert | **130.25** | 120.9 | 121.3 | NA |
| | Medium replay | **100.275** | 88.4 | 88.3 | NA |
| | Medium | **107.17** | 94.2 | 96.1 | NA |
| **HalfCheetah** | Expert | **95.52** | 93.6 | 90.5 | 94.5 |
| | Medium expert | 88.315 | 83.5 | 84.1 | **92.9** |
| | Medium replay | 43.58 | 44.3 | **44.8** | 47.2 |
| | Medium | 49.255 | 48.3 | 49.1 | **51.6** |
| **Hopper** | Expert | 103.8 | 103.4 | 88.1 | **110.6** |
| | Medium expert | **109.8** | 93.8 | 60.2 | 109.3 |
| | Medium replay | 92.995 | 92.1 | 63.1 | **103.2** |
| | Medium | 64.42 | 60.3 | 59.2 | **74.6** |
| **Walker2D** | Expert | 112.4 | 110.6 | 103.9 | **114.8** |
| | Medium expert | **112.3** | 109.0 | 96.5 | 110.1 |
| | Medium replay | 83.44 | 69.8 | 65.3 | **93.2** |
| | Medium | 82.6 | 82.7 | 71.3 | **85.2** |

### 3.1.3 Discrete environments

In discrete control tasks, we use implementations of PPO (Schulman et al., 2017b), DQN from StableBaselines3 (Raffin et al., 2021) to collect data. The mixed dataset has 2k (4%) near-optimal transitions and 48k (96%) transitions collected with a randomly initialized policy. We run INAC on Acrobot, Lunar Lander and Mountain Car on these datasets using the corresponding optimal hyper-parameters. We train for 70k iterations. We use learning rate of 0.0003. The results are displayed in Table 3.

### 3.2 Imbalanced Datasets

Most offline reinforcement learning (RL) algorithms return a target policy maximizing a trade-off between (1) the expected performance gain over the behavior policy that collected the dataset, and (2) the risk stemming from the out-of-distribution sampling of the induced state-action occupancy. It follows that the performance of the target policy is strongly related to the performance of the behavior policy. Therefore, it tends to generate trajectories that belong to a distribution similar to the distribution of the behaviour policy. We try with 5% and 50% "good" (having a higher reward) trajectories, using the datasets provided by Hong et al. (2023) for continuous control environments. The results for 5% good trajectory dataset are displayed in Table 4, and the results for 50% good trajectory dataset are displayed in Table 5.

Table 2: The final performance of each algorithm in continuous action space environments for expert, medium expert, medium replay and medium datasets. This table reports the score averaged over 6 random seeds.

| Environment | Dataset | INAC (Ours) | INAC | IQL | BCQ | TD3+BC |
|---|---|---|---|---|---|---|
| **Ant** | Expert | 4935.2 | **5077.9** | 4964.6 | 4907.4 | 2313.8 |
| | Medium expert | **5148.2** | 4750.2 | 4745.0 | 3825.2 | 3574.4 |
| | Medium replay | **3717.3** | 3391.0 | 2238.5 | 777.2 | 2986.2 |
| | Medium | 3623.4 | 3637.8 | **4006.4** | 2390.0 | 3594.9 |
| **HalfCheetah** | Expert | 11545.9 | 11347.0 | 11474.6 | 9243.8 | **12937.5** |
| | Medium expert | 10839.0 | 10086.4 | 7224.9 | 8674.7 | **11369.3** |
| | Medium replay | 5115.4 | 5209.8 | **5294.2** | 4005.5 | 4832.2 |
| | Medium | **5806.3** | 5716.7 | 5543.0 | 4693.2 | 5042.8 |
| **Hopper** | Expert | 3236.5 | 3346.4 | 2959.0 | **3347.9** | 2722.0 |
| | Medium expert | **3465.1** | 3032.0 | 758.8 | 1376.9 | 2731.4 |
| | Medium replay | 2659.6 | **2975.2** | 2013.7 | 52.8 | 768.8 |
| | Medium | 2139.9 | 1945.6 | 2018.3 | 1436.5 | **2415.6** |
| **Walker2D** | Expert | **5163.3** | 5076.3 | 5039.0 | 3347.9 | 4724.9 |
| | Medium expert | **5104.9** | 5006.6 | 4813.2 | 2455.1 | 4877.6 |
| | Medium replay | **3739.6** | 3205.2 | 3446.2 | 1600.9 | 605.9 |
| | Medium | **3824.3** | 3790.8 | 3801.6 | 3380.0 | 3566.5 |

Table 3: The final performance of each algorithm in discrete action space environments for optimal and mixed datasets. This table reports return per episode averaged over 6 random seeds.

| Environment | Dataset | INAC (Ours) | INAC | Oracle-Max |
|---|---|---|---|---|
| Acrobot | Opt | **-74.8** | -85.1 | -81.2 |
| | Mixed | -123.9 | **-91.0** | -146.8 |
| Lunar Lander | Opt | **294.0** | 201.2 | 166.5 |
| | Mixed | -468.2 | -255.4 | **-248.6** |
| Mountain Car | Opt | -207.8 | **-118.1** | -125.9 |
| | Mixed | -1000.0 | **-151.2** | -187.9 |

### 3.3 Comparison with other in-sample algorithms

We compare the in-sample offline reinforcement algorithms - INAC, IAC, IQL and BCQ. The differentiating concepts in each algorithm are as follows: INAC implements in-sample softmax, IQl treats the state value function as a random variable and then takes its state conditional upper expectile to estimate the best actions in that state, IAC implements sample-importance-resampling (Rubin & Rubin, 1988), and BCQ implements agents which are trained to maximize reward while minimizing the mismatch between the state-action visitation of the policy and the state-action pairs contained in the batch. INAC and IAC gave comparable results, and were significantly better than BCQ. The returns of INAC in different continuous control environments are compared with the returns of BCQ, IQL, and TD3+BC in Figure 2.

### 3.4 Behaviour Cloning (BC) Regularization

We observe the effect of adding Behaviour Cloning Regularization (Fujimoto & Gu, 2021) along with entropy regularization (Mnih et al., 2016). This results in the following loss function for actor $\pi_\psi$

$$\mathcal{L}_{actor}(\psi) = -\mathbb{E}_{s,a\sim\mathcal{D}}\left[\exp\left(\frac{q_\theta - v_\psi(s)}{\tau} - \log \pi_\omega(a|s)\right)\log \pi_\psi(a|s) \cdot \lambda - (\pi_\psi(s) - a)^2\right] \quad (8)$$

Table 4: Performance comparison of each algorithm in continuous action space environments for imbalanced datasets - expert and medium for mixing ratio 0.05. Performance was averaged over 3 random seeds.

| Environment | Dataset | INAC (Ours) | IQL | CQL |
|---|---|---|---|---|
| Ant | Expert | 111.6 | 105.4 | **115.5** |
| | Medium | 89.7 | 89.3 | **90.2** |
| HalfCheetah | Expert | 64.3 | 55.4 | **69.7** |
| | Medium | 43.5 | **47.3** | 41.5 |
| Hopper | Expert | 108.4 | **110.6** | 107.3 |
| | Medium | 42.7 | 59.8 | **63.1** |
| Walker2D | Expert | 107.1 | 104.3 | **108.3** |
| | Medium | 71.3 | 67.9 | **75.2** |

Table 5: Performance comparison of each algorithm in continuous action space environments for imbalanced datasets - expert and medium for mixing ratio 0.5. Performance was averaged over 3 random seeds.

| Environment | Dataset | INAC (Ours) | IQL | CQL |
|---|---|---|---|---|
| Ant | Expert | 111.6 | 130.3 | **132.7** |
| | Medium | 89.7 | 98.9 | 97.4 |
| HalfCheetah | Expert | 64.3 | **94.4** | 86.3 |
| | Medium | 43.5 | **47.1** | 46.5 |
| Hopper | Expert | 108.4 | **109.4** | 106.7 |
| | Medium | 42.7 | 61.9 | **64.5** |
| Walker2D | Expert | 107.1 | **109.4** | 75.6 |
| | Medium | 71.3 | 71.2 | **89.1** |

Incorporating the highlighted term introduces BC regularization. Here, $\lambda$ denotes the hyperparameter governing the regularization intensity. We set $\lambda$ to 0.1 consistently across all instances employing BC Regularization. Adding this explicit regularization to the actions taken in the dataset increases the stability during training while keeping the mean rewards the same in datasets with significant amounts of good trajectories. As quality of the dataset reduces, this regularisation contributes to a decrease in mean return. The results and the graphs are visualized in Figure 3 and in the Appendix, Figure 7.

## 4 Discussion

### 4.1 Uniqueness of INAC

Our results agree with the original authors' claims. INAC steers clear of depending on actions drawn from an approximation of $\pi_{\mathcal{D}}$, instead employing expectile regression. Previous approaches, such as BCQ, aimed to devise a straightforward algorithm based on an in-sample max algorithm but required the integration of multiple techniques. Apart from the fact that the algorithm completely avoids out-of-distribution actions, in-sample softmax's strengths lie in the application of the softmax function in place of a hardmax. A softmax keeps information about the non-maximal elements intact (which is something that a hardmax does not), and this leads to a much better approximation.

### 4.2 Performance dependence on dataset

Due to the dependency on behavior policy performance, most offline RL algorithms are susceptible to the return distribution of the trajectories in the dataset collected by a behavior policy. In a near-optimal

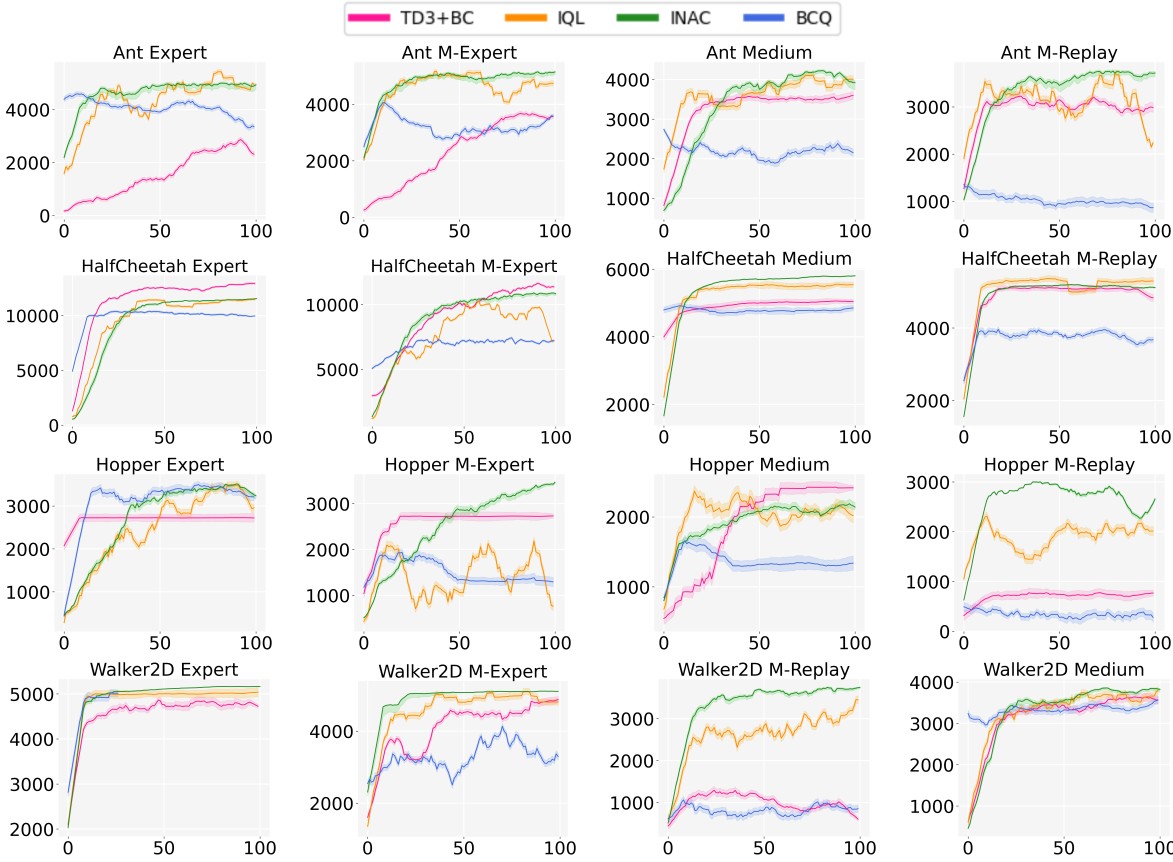

Figure 2: Average Returns vs Iterations (x10$^4$). INAC over performs or is comparable to other in-sample algorithms like BCQ or IQL across all datasets and all environments

dataset (i.e., expert) this tendency favors the performance of an algorithm, while in a low-performing dataset often reduces the policy's performance. In realistic scenarios, offline RL datasets might consist mostly of low-performing trajectories with few minor high-performing trajectories collected by a mixture of behavior policies since collecting high-performing trajectories is costly and difficult. It is thus desirable to avoid anchoring on low-performing behavior policies and exploit high-performing ones in mixed datasets.

## 4.3 INAC compared to other algorithms

In such imbalanced datasets, INAC performs comparable to variations of IQL and CQL, devised using Return Weighting and advantage weighting sampling strategies as proposed by Hong et al. (2023). The distribution learned due to the in-sample softmax approach allows INAC to learn the high-performing transitions in data distribution effectively, thus allowing it to perform comparably to methods devised explicitly for the same. We verify that learning an in-sample softmax performs much better in comparison to learning the in-sample max. INAC performs much better than BCQ in the D4RL continuous environments. We can further infer that INAC effectively learns a policy which is close to the optimal policy even in expert settings.

## 4.4 Effect of BC Regularization

When incorporating BC Regularization, we observe a stable training run without a significant change in the mean return in all runs in expert, and medium expert datasets. This is because adding explicit BC regularisation will decrease performance, when quality of actions in the dataset reduces. This phenomenon suggests that explicitly guiding the policy to emulate expert actions from the dataset may alter the policy's distribution while not necessarily enhancing its overall performance. These findings suggest that INAC's

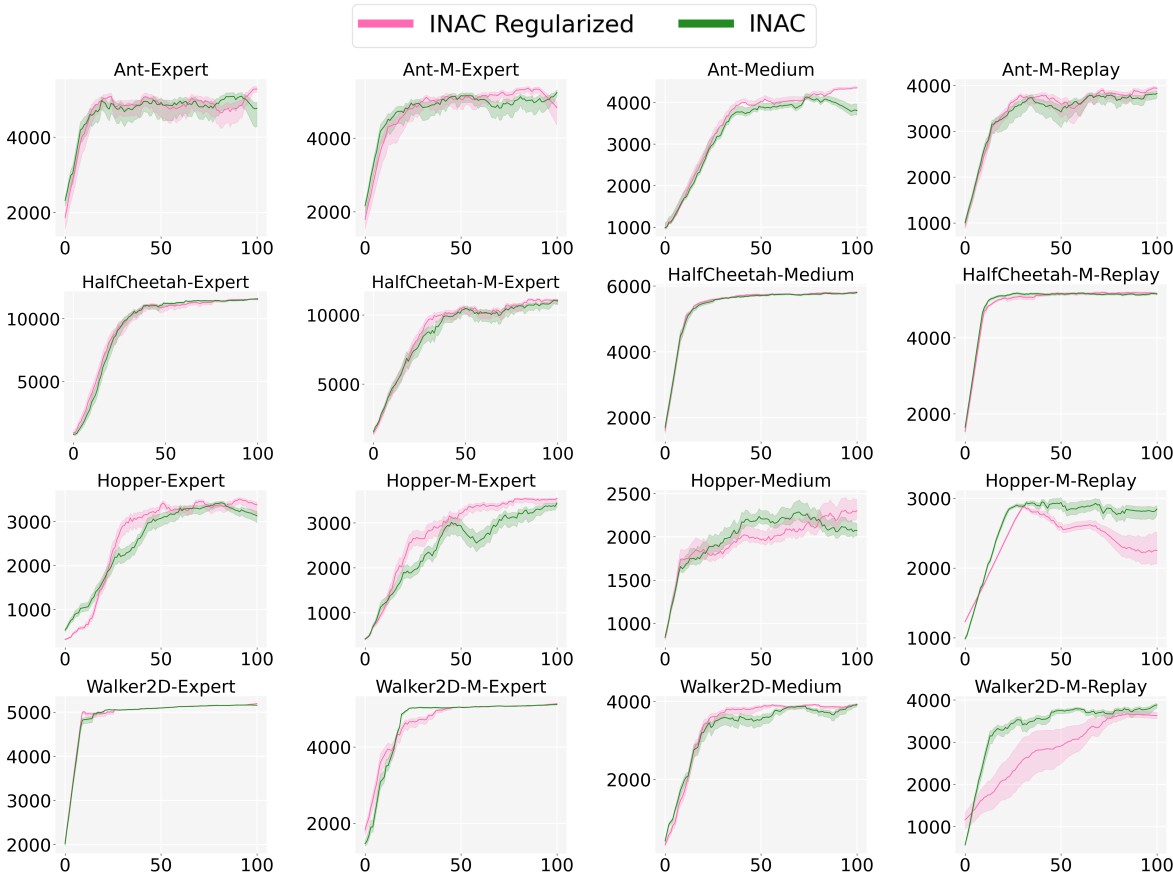

Figure 3: Average Returns vs Iterations (x10$^4$). Performance was averaged over 3 random seeds.

inherent capability to learn a robust action distribution renders further regularization unnecessary. Since there isn't a significant improvement with regularization in expert settings, INAC learns close to a perfect policy.

## 5 Conclusion

Our attempt to reproduce the findings of Xiao et al. (2023) regarding the In-sample Softmax Algorithm yielded results consistent with theirs across all three environments tested: tabular, continuous, and discrete. In the domain of offline RL with imbalanced datasets, the prevailing state-of-the-art approaches are Conservative Q-Learning and Implicit Q-Learning, which employ the re-weighted sampling strategy introduced by Hong et al. (2023). Our investigation finds that INAC performs similarly to these algorithms in continuous control environments. This suggests that INAC can be directly applied when datasets are imbalanced. Our evaluation against Batch Conservative Q-Learning demonstrates that INAC surpasses BCQ by a considerable margin and achieves performance similar to IAC across all continuous control environments. Therefore, we conclude that INAC exhibits performance on par with state-of-the-art algorithms across diverse settings and shows resilience to changes in training data distributions. We find that INAC's data-agnostic nature makes it a reliable and safe choice for offline RL, particularly in settings with unknown data distributions.

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

# A Appendix

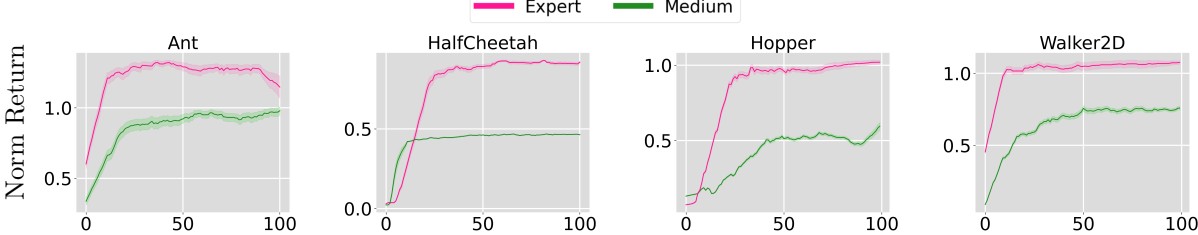

Figure 4: Normalized Average Returns vs Iterations (x$10^4$). The performance of INAC for expert and medium environments provided an imbalanced dataset comprising of 50% good and 50% bad trajectories. Performance was averaged over 3 random seeds, after using a smoothing window of size 15

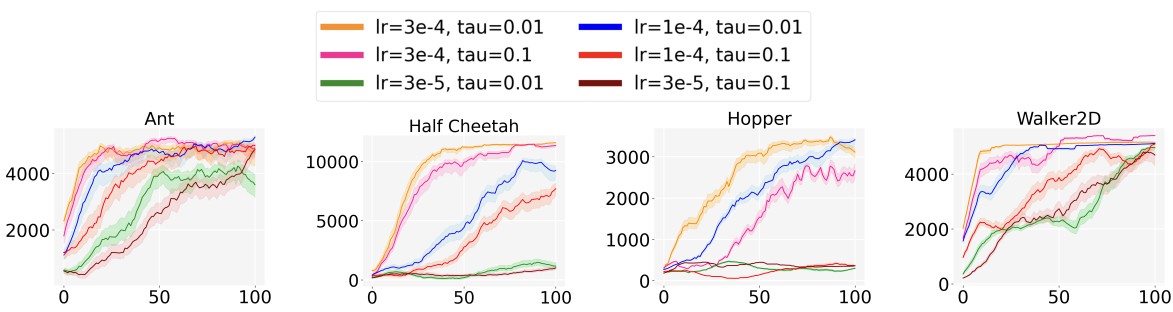

Figure 5: Average Returns vs Iterations (x$10^4$). INAC performance comparison of different hyperparameters in different environments. The graph infers that the hyperparameters given in the original paper were optimal. Performance was averaged over 3 random seeds, after using a smoothing window of size 15

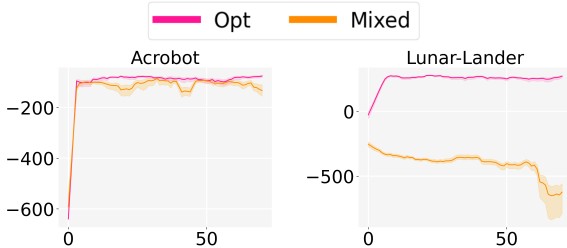

Figure 6: Average Returns vs Iterations (x$10^3$). The average returns for INAC on discrete environments have been reproduced. Performance was averaged over 3 random seeds, after using a smoothing window of size 15

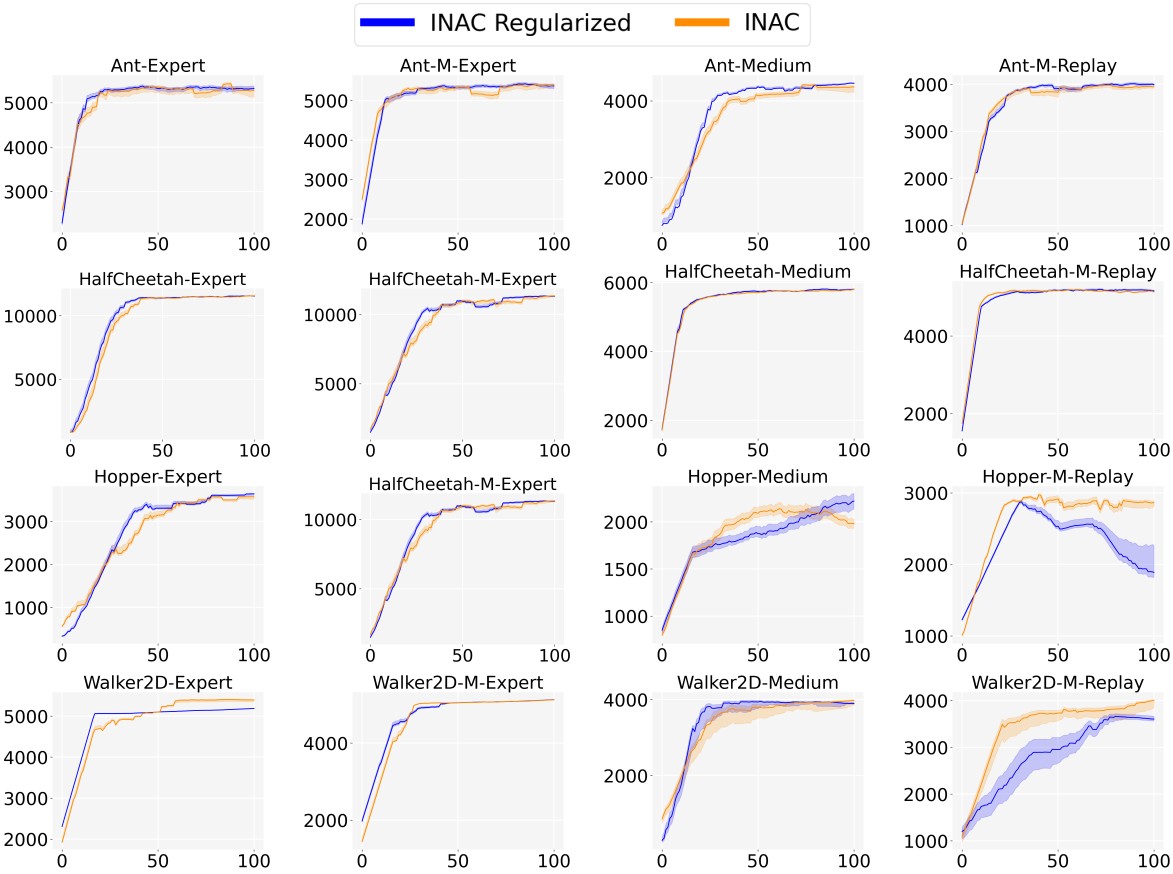

Figure 7: Median Returns vs Iterations (x10$^4$). BC regularization pushes the distribution further towards the dataset, leading to better stability of the median return in expert and medium expert datasets. Performance was averaged over 3 random seeds, after using a smoothing window of size 15

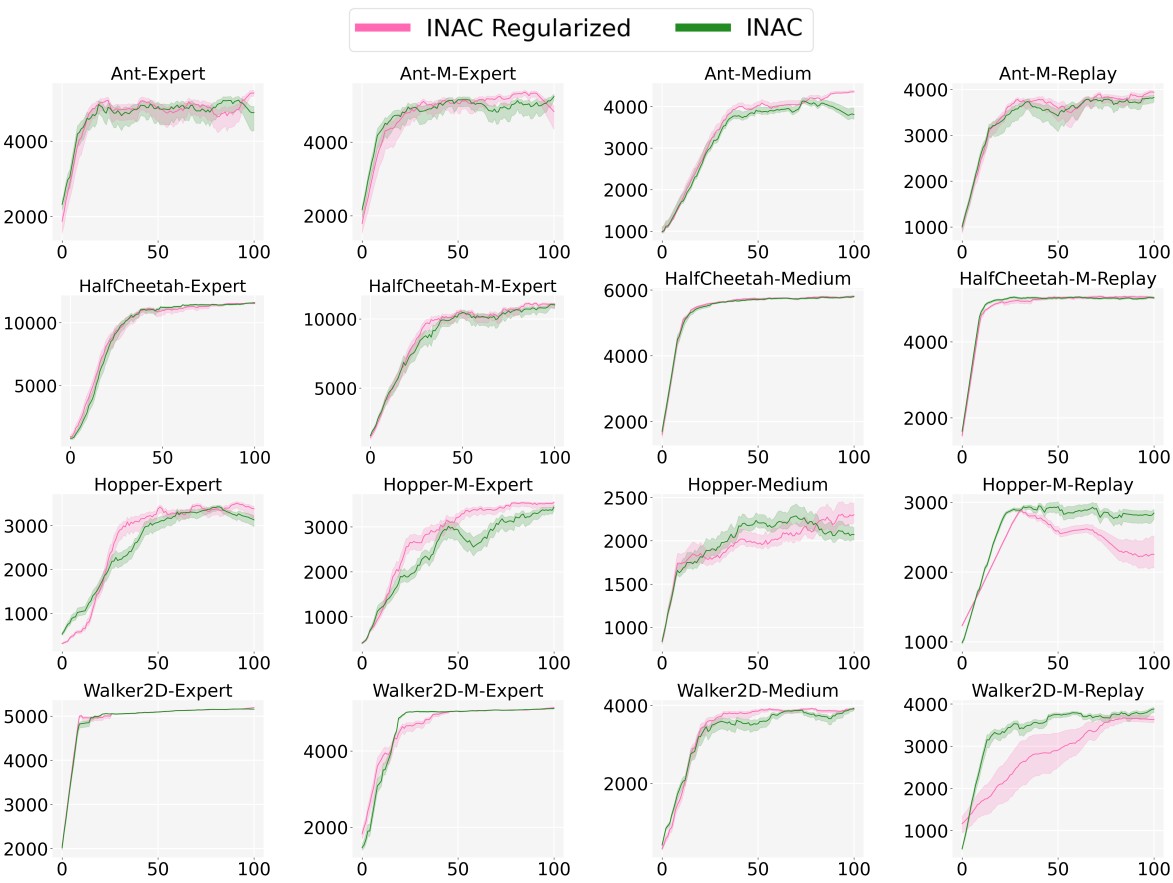

Figure 8: Average Returns vs Iterations (x10$^4$). BC regularization added to INAC improves the average return, especially in the expert and medium expert datasets as expected. Performance was averaged over 3 random seeds, after using a smoothing window of size 15

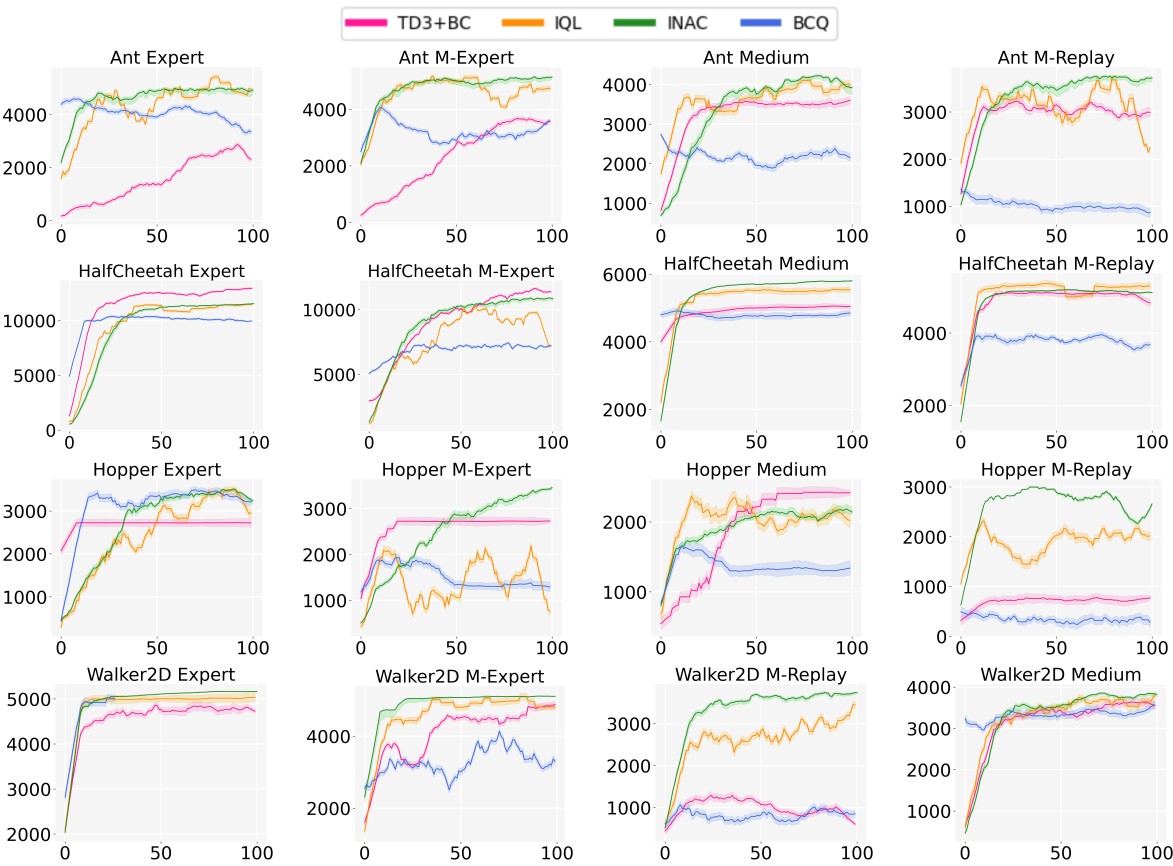

Figure 9: Average Returns vs Iterations (x10$^4$). INAC over performs or is comparable to BCQ across all datasets and all environments.

Table 6: The final performance of each algorithm in continuous action space environments for expert, medium expert, medium replay and medium datasets. This table reports the score before normalization. Performance was averaged over 6 random seeds.

| Environment | Dataset | INAC (Ours) | INAC | IQL | BCQ | TD3+BC |
|---|---|---|---|---|---|---|
| **Ant** | Expert | 4935.2 | **5077.9** | 4964.6 | 4907.4 | 2313.8 |
| | Medium expert | **5148.2** | 4750.2 | 4745.0 | 3825.2 | 3574.4 |
| | Medium replay | **3717.3** | 3391.0 | 2238.5 | 777.2 | 2986.2 |
| | Medium | 3623.4 | 3637.8 | **4006.4** | 2390.0 | 3594.9 |
| **HalfCheetah** | Expert | 11545.9 | 11347.0 | 11474.6 | 9243.8 | **12937.5** |
| | Medium expert | 10839.0 | 10086.4 | 7224.9 | 8674.7 | **11369.3** |
| | Medium replay | 5115.4 | 5209.8 | **5294.2** | 4005.5 | 4832.2 |
| | Medium | **5806.3** | 5716.7 | 5543.0 | 4693.2 | 5042.8 |
| **Hopper** | Expert | 3236.5 | 3346.4 | 2959.0 | **3347.9** | 2722.0 |
| | Medium expert | **3465.1** | 3032.0 | 758.8 | 1376.9 | 2731.4 |
| | Medium replay | 2659.6 | **2975.2** | 2013.7 | 52.8 | 768.8 |
| | Medium | 2139.9 | 1945.6 | 2018.3 | 1436.5 | **2415.6** |
| **Walker2D** | Expert | **5163.3** | 5076.3 | 5039.0 | 3347.9 | 4724.9 |
| | Medium expert | **5104.9** | 5006.6 | 4813.2 | 2455.1 | 4877.6 |
| | Medium replay | **3739.6** | 3205.2 | 3446.2 | 1600.9 | 605.9 |
| | Medium | **3824.3** | 3790.8 | 3801.6 | 3380.0 | 3566.5 |

Table 7: The final performance of each algorithm in continuous action space environments for expert, medium expert, medium replay and medium datasets. Scores are normalized. Performance was averaged over 6 random seeds.

| Environment | Dataset | INAC (Ours) | INAC | IQL | IAC |
|---|---|---|---|---|---|
| **Ant** | Expert | **126.7** | 128.4 | 118.9 | NA |
| | Medium expert | **130.25** | 120.9 | 121.3 | NA |
| | Medium replay | **100.275** | 88.4 | 88.3 | NA |
| | Medium | **107.17** | 94.2 | 96.1 | NA |
| **HalfCheetah** | Expert | **95.52** | 93.6 | 90.5 | 94.5 |
| | Medium expert | 88.315 | 83.5 | 84.1 | **92.9** |
| | Medium replay | 43.58 | 44.3 | **44.8** | 47.2 |
| | Medium | 49.255 | 48.3 | 49.1 | **51.6** |
| **Hopper** | Expert | 103.8 | 103.4 | 88.1 | **110.6** |
| | Medium expert | **109.8** | 93.8 | 60.2 | 109.3 |
| | Medium replay | 92.995 | 92.1 | 63.1 | **103.2** |
| | Medium | 64.42 | 60.3 | 59.2 | **74.6** |
| **Walker2D** | Expert | 112.4 | 110.6 | 103.9 | **114.8** |
| | Medium expert | **112.3** | 109.0 | 96.5 | 110.1 |
| | Medium replay | 83.44 | 69.8 | 65.3 | **93.2** |
| | Medium | 82.6 | 82.7 | 71.3 | **85.2** |

