# OpenReview forum: "Evaluating In-Sample Softmax in Offline Reinforcement Learning: An Analysis Across Diverse Environments"
_TMLR — Rejected by TMLR_

### Review · Reviewer_eWmc · 2024-03-10

**Summary Of Contributions:**

This study investigates the In-Sample Softmax (INAC) algorithm for Offline RL, focusing on learning from fixed datasets with incomplete action coverage. It compares INAC to similar algorithms across various environments, revealing its robust performance and competitive advantages. The analysis underscores INAC's potential in addressing offline RL challenges.

**Audience:**

No

**Claims And Evidence:**

Yes

**Requested Changes:**

1. Perhaps the authors can write a more extensive comparison on why the experiments conducted in this manuscript are different from the original INAC paper. Either in a literature review section or in the preliminary section.
2. Once the previous point is added, the authors can try to add more empirical comparisons.

**Strengths And Weaknesses:**

Strength:
1. This paper is clearly written and easy to follow.

Weakness:
1. It is hard to tell how it is different from the original INAC paper – since the original INAC paper also has comparison experiments in the OOD problem while this manuscript also studies the robustness of INAC. As far as the reviewer can tell, this manuscript focuses on reproducing the results with previous offline RL methods, which does not appear as a typical research paper.
2. Lack of empirical comparison. The INAC paper contains comparison experiments with CQL, and TD3 + BC, while this submission contains fewer comparison experiments.
3. Missing literature review section – it seems that this manuscript does not contain a literature review section.

---

> ### Author Response · Authors · 2024-05-07
> **Reply to Reviewer eWmc**
>
> Thank you for your review. TMLR does accept both reproducibility and replicability studies, which are important for ensuring the quality and integrity of scientific research. Having run additional experiments that go beyond simple verification of the original authors’ experiments, we believe this paper is a valuable addition to reproducibility research in offline RL. The additional experiments include running InAC with Behaviour Cloning regularization along with entropy regularization. The conclusions from InAC with BC regularization have been included in the paper in section 4.4, which is that even after incorporating BC regularization, we observe a stable training run without a significant change in the mean returns in the expert and medium-expert datasets. This proves that InAC learns a policy comparable to the ideal policy, showcasing how it indeed is a state-of-the-art offline RL algorithm. We also ran InAC on imbalanced datasets, to further prove the robustness of this algorithm. Furthermore, we have included the empirical comparisons with TD3+BC and IQL in the revised version of our paper. We have also updated our background section to include the necessary preliminaries, which should give a clear idea about the literature review.
>
> Thank you for the review. We have implemented most of the changes suggested. We feel your reviews and comments have had a significantly positive impact on improving the quality of our work.

---

### Review · Reviewer_Zwy3 · 2024-03-19

**Summary Of Contributions:**

The paper studies in-sample actor-critic algorithm for offline RL evaluated on a wide range of benchmark tasks. The aim of the study is to benchmark the performance of in-sample actor-critic in a reproducible way, and assess its improvement over prior methods such as BCQ and IQL.

**Audience:**

Yes

**Claims And Evidence:**

Yes

**Requested Changes:**

=== Eqn 4 ===

It's been argued towards the end of page 3 that INAC learns $\pi_w\approx \pi_D$, but in Eqn 4, $\pi_D$ and $\pi_w$ appear both on the RHS to define the target policy. In case $\pi_w=\pi_D$, would that be the case that they cancel out on the RHS of Eqn 4? What's the motivation for having both $\pi_D$ and $\pi_w$ on the RHS of Eqn 4 to define the target policy?

In case $\pi_w$ is learned well, which is what the policy intends to do, it will recover $\pi_D$ as the optimal solution (based on the MLE objective). Wouldn't that mean the target softmax policy defined through $q_\theta$, and has no regularization towards any policy?

=== Eqn 8 ===

In Eqn 8, there is a behavior regularization term motivated by BCQ added to the INAC loss function. The authors have mentioned that adding this regularization with $\lambda=0.1$ tends to stabilize training - would that mean that the original objective as defined in Eqn 4 is somehow incomplete, in that it is not enough to guarantee stable training from offline dataset. In hindsight it seems to make sense since Eqn 4 yields an optimal policy that is only defined through $q_\theta$ and not regularized against anything, hence an extra regularization loss might be necessary.

How is the coefficient $\lambda$ chosen in practice?

==== Tables ===

Across all experiments, it's been reported that the mean values are averaged across 3 random seeds. I think as good reporting practice, one should also report the standard deviations across runs, and increase the number of runs from 3 to higher if budget allows. Since the main point of the paper is reproducibility, reporting only mean performance across 3 seeds is not a gold standard for reproducible research.

=== Fig 2 ===

Fig 2 shows the training performance of BCQ vs. INAC over time. How about other algorithms such as IQL, which is also an important baseline? We have seen its numbers in tables, but should expect to see its training curves in Fig 2 too?

It seems that the paper has placed lots of emphasis on comparison against BCQ. Why would this be the case? In the INAC formulation there is little mention of behavior regularization, which is the defining characteristic of BCQ. Meanwhile, INAC has a regularization version which takes inspirations from BCQ. It feels that a direct comparison against BCQ would not be as informative since the two methods are quite orthogonal with one another.

=== Presentation ===

The paper should ideally also present better the background info on BCQ and IQL. In its current form every algorithm is described in word and it is hard to tell the connection / difference between these algorithms.

**Strengths And Weaknesses:**

The attempt to carry out reproducible research is good especially for the RL community, where reproducibility of results has been a long-standing issue. The evaluation carried out by the paper looks relatively comprehensive, and can be served as reference practices for future work.

In the meantime, the paper does not provide much novelty in terms of technical contributions to the RL research. The algorithms are already existent and it is not clear if the experiments in the paper provide significant new insights as to how offline RL works, and why certain algorithms fail.

---

> ### Author Response · Authors · 2024-05-07
> **Reply to Reviewer Zwy3**
>
> > In the meantime, the paper does not provide much novelty in terms of technical contributions to the RL research...
>
> Thank you for your review. By studying the performance of the  InAC algorithm across diverse environments and also against other state-of-the-art offline RL algorithms, we aimed to verify the original authors’ claim that InAC is indeed a suitable algorithm for any problem that requires offline reinforcement learning. The novelty of our paper is that InAC’s performance was tested with Behaviour Cloning regularization, the significance of which has been elaborated in section 4.4, which is that even after incorporating BC regularization, we observe a stable training run without a significant change in the mean returns in the expert and medium-expert datasets. This proves that InAC learns a policy comparable to the ideal policy, showcasing how it indeed is a state-of-the-art offline RL algorithm. Apart from this we also tested InAC on imbalanced datasets provided by [1], something that the original authors had not run experiments on.
>
> > === Eqn 4 ===
>
> To elaborate on the significance of equation 4, we would like to mention that the algorithm essentially tries to approach a target policy with its distribution as a softmax over the actions, considering the Q-function as a quantifier. If this were only max, the ideal policy would give a probability of 1 at the maximum Q-value. Here in the equation, the policy is expressed so because $\pi_D$ can be considered the implicit behaviour policy for the dataset we use. While we have access to the dataset, we cannot access the exact state action distribution. Hence, we approximate the same with $\pi_\omega$. Of course, when the approximation turns out to be the ideal case, that is, $\pi_\omega$ = $\pi_D$, they would cancel out, leaving only the softmax term on the RHS, which is as expected.
>
> > === Eqn 8 ===
>
> Thank you for drawing attention to this crucial point. We would like to start by saying that yes, the policy given in equation 4 is not explicitly regularized. It just restricts the state action pairs used for bellman backups to lie within the dataset. This explicit regularization has empirically shown to stabilize the training in our experiments. $\lambda$ is originally defined as the inverse of the average of the absolute values of $Q(s, a)$ as given by [2].
>
> > === Tables ===
>
> Thank you for the suggestion. We have now added the results of the additional seeds and baselines that we experimented with.
>
> > === Fig 2 ===
>
> We have chosen to compare BCQ with InAC because BCQ is an algorithm that uses an in-sample argmax function to update its value function. We aimed to verify the original authors’ claims that using an in-sample softmax is an inherently better approach than employing an in-sample argmax. Having gotten substantially better returns in InAC, we have obtained empirical evidence to support this claim.  Apart from this, we have revised our initial submission to also include graphs for our runs of IQL and TD3+BC as well.
>
> > === Presentation ===
>
> We have revised the background section of our paper and included necessary preliminaries to improve the overall clarity.
>
> Thank you for an incredibly detailed review. We have implemented most of the changes suggested. We feel your reviews and comments have had a significantly positive impact on improving the quality of our work.
>
> [1] Zhang-Wei Hong, Pulkit Agrawal, Rémi Tachet des Combes, and Romain Laroche. Harnessing mixed offline reinforcement learning datasets via trajectory weighting, 2023.
>
> [2] Scott Fujimoto and Shixiang Shane Gu. A minimalist approach to offline reinforcement learning, 2021.

---

### Review · Reviewer_nYuz · 2024-04-22

**Summary Of Contributions:**

This paper reproduces the findings of Xiao et al. (2023) on the InAC algorithm across various environments. It also compares InAC with popular offline RL approaches like BCQ, IQL, and IAC. The experimental results on D4RL confirm that InAC is a reliable offline RL algorithm, showcasing comparable or superior performance compared to existing methods. A study on imbalanced dataset and behavior cloning regularization for InAC is also provided.

**Audience:**

Yes

**Claims And Evidence:**

Yes

**Requested Changes:**

**Major Comments**

1. In Section 2.2, I recommend placing greater emphasis on delineating the significant advantages of in-sample policy optimization, particularly in comparison with KL-regularization methods. Providing a detailed explanation of why in-sample optimization methods are superior in avoiding the selection of out-of-distribution actions would strengthen the motivation for conducting the reproducibility of in-sample policy optimization algorithms. Moreover, these sentences are directly extracted from the penultimate paragraph of Section 2.2 in the paper [Xiao et al., 2023].
2. In Section 2.5, the first sentence asserts that “The in-sample AC algorithm [Xiao et al., 2023] carefully considers out-of-sample actions.” However, it is unclear from the provided information how exactly the in-sample AC algorithm accomplishes this. I suggest either expanding on how the algorithm addresses out-of-sample actions or considering removal of this statement.
3. This is an experimental research paper for reproducibility study. I am concerned that using only 3 seeds for the experiments might not provide sufficient evidence to persuade readers of the convincingness and validity of the reproduced results. Increasing the number of seeds could enhance the robustness and reliability of the findings. Moreover, additional analysis on the statistical significance would make the results in this paper more convincing.

**Other Comments**

1. On page 3, in the last sentence of Section 2.2, there should be a space between the comma and “may” in the phrase “approach, may not avoid…”.
2. Do the dots in Equation (2) need to be represented as \cdot? I'm uncertain whether this is a minor issue or a problem with the TMLR template.
3. The title of Section 2.5 requires a space between “Softmax” and “(INAC).” Similar issues also occur in multiple places.
4. While Section 2.3 and 2.4 are understandable to those who already have a good idea about BCQ and IAC, they could be quite hard to parse for readers who are not that familiar with these methods.
5. In Section 3.1.1 and later, it would be beneficial to provide a clear definition of the “Mixed dataset.” From the context, it seems that the 6. "Mixed dataset" refers to a combination of data from the Expert dataset and the Random dataset. Clarifying this terminology within the paper would enhance reader understanding and ensure transparency in the methodology.
7. I recommend highlighting the best performance result in each environment and configuration within the performance results tables by making its font bold. This enhancement would make it easier for readers to quickly identify the most significant outcomes in the presented tables.
8. There should be a period at the end of the last sentence in Section 3.1.3.
9. In Section 3.2, it would be advantageous to provide a precise definition of what constitutes "good" trajectories. Clarifying this terminology will ensure a clear understanding for readers and enhance the coherence of the section.
10. The exact update of InAC does not employ the $\hat{\pi}{\pi{\mathcal{D}}, q_{\theta}}$ as defined in (4). This in-sample soft greedy policy, detailed in the paper by Xiao et al. (2023) for the derivation of their algorithm, appears to be omitted in this context. Therefore, it may be advisable to either remove equation (4) if it is not utilized in the paper, or alternatively, retain it and provide further elucidation on the algorithmic derivations associated with (4).
11. There are many inconsistent terminologies and notations. It would be helpful to have another round of proofreading of this paper.

I find this paper lacking in quality as it only conducts a limited number of experiments (with 3 seeds) for reproducibility without delving deeper into understanding the InAC algorithm or any in-sample policy optimization algorithm. Furthermore, the paper's writing style is subpar as it relies heavily on directly pasting sentences from other papers, leading to logical inconsistencies and confusion among readers.

**Strengths And Weaknesses:**

**Strengths**

- This paper thoroughly conducts the reproduction of the InAC algorithm under several different environments under various settings, such as discrete/continuous environments, imbalanced datasets, and behavior cloning regularization.
- The experimental findings demonstrate that InAC achieves performance comparable to or better than several benchmark offline RL methods.

**Weaknesses**

- Most parts of the paper appear to be direct verification of the experimental results in the original InAC paper. It remains unclear if this paper contributes new insights or sufficient new value through additional analysis or ablations.
- In Section 4 (the Discussion section), while the paper effectively presents depictions of the experimental results, it would be valuable to delve deeper into the underlying reasons for the outstanding performance of InAC. Currently, Section 4 only reiterates the direct observations shown in the tables / figures in Section 3. Offering additional insights into the factors contributing to its efficacy would enhance the value of the paper for reproducibility research.
- **Numerous sentences in Section 2 appear to be directly copied verbatim from the original paper**, as evidenced by similarities to [Xiao et al., 2023]. Just to name a few: (i) Section 2.1; (ii) Section 2.2, the first two sentences; (iii) Section 2.2, Equation (2)-(3) and the description of these equations. (iv) Almost the whole Section 2.5. This is serious plagiarism and definitely not allowed.
- I believe there is substantial room for improvement in the paper presentation, particularly in the writing aspect. There are instances where several sentences convey the same idea, so merging them into one would enhance clarity and conciseness.
- Except Section 3.4, it would be helpful to conduct additional experiments integrating various regularization or combining InAC with optimization techniques to assess performance of InAC under diverse conditions.

---

> ### Author Response · Authors · 2024-05-07
> **Reply to Reviewer nYuz**
>
> > Most parts of the paper appear to be direct verification of the experimental results....
>
> This paper is a reproducibility study, of which there is a dearth, especially in the RL community, hence most parts of the paper indeed include verifying the author's claims. However, we believe that because of the additional experimentation conducted, i.e. the BC regularization experiments and the experiments with mixed datasets, this paper does provide value to RL research.
>
> > In Section 4 (the Discussion section), while the paper effectively presents depictions of the experimental results...
>
> Thank you for bringing attention to this crucial point.  We have included changes in section 4 which portray the underlying reasons for the outstanding performance of InAC. We would like to re-iterate though, that as we have stated in section 2.5, InAC does not consider out-of-distribution actions, which leads to the algorithm not needing to extrapolate for previously unseen actions, which often becomes a source of error in an offline RL algorithm. Implementing a softmax is inherently better than a hardmax because the softmax keeps the information of the non-maximal elements intact, which leads to a better approximation. This has also been empirically verified in our comparison of InAC with BCQ.
>
> > Numerous sentences in Section 2 appear to be directly copied verbatim from the original paper...
>
> We have made the changes in all of the necessary places in our paper. We used the original paper as a reference, but due to the concerns raised, we have rewritten the concerned parts.
>
> > I believe there is substantial room for improvement in the paper presentation, particularly in the writing aspect....
> > In Section 2.5, the first sentence asserts that “The in-sample AC algorithm [Xiao et al., 2023] carefully considers out-of-sample actions.”
>
> We have incorporated the necessary changes as suggested in our revision. We have gone through the paper once more and have hopefully eliminated the unnecessarily repetitive nature of our writing. Any repetitions included are deliberate to provide emphasis to those certain points. We have done a thorough grammatical review in our revision and have incorporated most of the minor changes suggested.
>
> > In Section 2.2, I recommend placing greater emphasis on delineating the significant advantages...
>
> Thank you for bringing this to our notice. We have appropriately changed section 2.2 and included a brief explanation as to why the in-sample policy optimisation strategy has an advantage over the KL divergence strategy when considering in-sample actions.
>
> > Except Section 3.4, it would be helpful to conduct additional experiments integrating...
> > This is an experimental research paper for reproducibility study. I am concerned that using only 3 seeds...
>
> We appreciate your point highlighting the need for more seeds and accordingly, we have increased the number of seeds from 3 to 6. We hope that our paper now provides sufficient evidence to persuade readers of the convincingness and validity of the reproduced results. We ran InAC against additional baselines like IQL, BCQ, and TD3+BC to compare its performance against these standard algorithms, and conducted experiments going beyond the original paper to further boost the validity of this study. We would like to reiterate that we are operating under severe computational constraints.
>
> Thank you for an incredibly detailed review. We have implemented most of the changes suggested and we have also rectified all the grammatical errors. We feel your reviews and comments have had a significantly positive impact on improving the quality of our work.

---

### Author Response · Authors · 2024-05-07
**General reply to Reviewers**

We would like to thank all of the reviewers for their insightful and thoughtful comments. To address some common concerns, this paper is primarily a reproducibility study of the In-Sample Softmax algorithm for offline reinforcement learning. We have verified the original authors’ claims as well as run additional experiments. This paper serves as a study on the robustness of InAC across diverse environments. We ran a sanity check, and experiments with and without Behaviour Cloning regularization along with entropy regularization on four continuous MuJoCo environments using four types of D4RL datasets, three discrete MuJoCo environments using two D4RL datasets, and four continuous MuJoCo environments using two types of mixed datasets provided by [1]. Whilst operating under computational constraints, we have run these experiments with the maximum number of seeds feasible, and have even increased the number of seeds after the initial submission in some places. We hope we have answered all the other specific queries below.


[1] Zhang-Wei Hong, Pulkit Agrawal, Rémi Tachet des Combes, and Romain Laroche. Harnessing mixed offline reinforcement learning datasets via trajectory weighting, 2023.

---

### Decision · Action_Editor_mf18 · 2024-06-10

**Recommendation:** Reject

**Comment:**

After rebuttal, unfortunately, none of the reviewers are in favor of this reproduction study. While good reproduction studies do fall into the scope of TMLR, the current paper focuses mostly on simple verification, and falls short of contributing sufficient new perspectives through additional analysis or ablations. I encourage the authors to go beyond simple verifications and try to identify new intuitions / perspectives, or even potential ideas for improved algorithms, which could significantly strengthen the paper in my opinion.

**Audience:**

The topic of the paper could be of interest to the RL community; however, the reproduction results at the current stage may not sound interesting enough to this audience.

**Claims And Evidence:**

This paper provides a reproduction study of the In-Sample Softmax (INAC) algorithm of Xiao et al. (2023) for offline reinforcement learning. The experiments verify that INAC performs similarly as other state-of-the-art algorithms across many settings, and identifies some concrete cases where INAC is superior. Main claims of the paper are supported by these experiments.

**Resubmission Of Major Revision:**

The authors may consider submitting a major revision at a later time.